# Causal Associations of Glaucoma and Age-Related Macular Degeneration with Cataract: A Bidirectional Two-Sample Mendelian Randomisation Study

**DOI:** 10.3390/genes15040413

**Published:** 2024-03-26

**Authors:** Je Hyun Seo, Young Lee

**Affiliations:** 1Veterans Medical Research Institute, Veterans Health Service Medical Center, Seoul 05368, Republic of Korea; lyou7688@bohun.or.kr; 2Department of Applied Statistics, Chung-Ang University, Seoul 06974, Republic of Korea

**Keywords:** cataract, primary open-angle glaucoma, age-related macular degeneration, Mendelian randomisation, single-nucleotide polymorphisms

## Abstract

Common age-related eye disorders include glaucoma, cataract, and age-related macular degeneration (AMD); however, little is known about their relationship with age. This study investigated the potential causal relationship between glaucoma and AMD with cataract using genetic data from multi-ethnic populations. Single-nucleotide polymorphisms (SNPs) associated with exposure to cataract were selected as instrumental variables (IVs) from genome-wide association studies using meta-analysis data from BioBank Japan and UK Biobank. A bidirectional two-sample Mendelian randomisation (MR) study was conducted to assess the causal estimates using inverse variance weighted, MR-Egger, and MR pleiotropy residual sum and outlier tests. SNPs with (*p* < 5.0 × 10^−8^) were selected as IVs for cataract, primary open-angle glaucoma, and AMD. We found no causal effects of cataract on glaucoma or AMD (all *p* > 0.05). Furthermore, there were no causal effects of AMD on cataract (odds ratio [OR] = 1.02, *p* = 0.400). However, glaucoma had a substantial causal effect on cataract (OR = 1.14, *p* = 0.020). Our study found no evidence for a causal relationship of cataract on glaucoma or AMD and a casual effect of AMD on cataract. Nonetheless, glaucoma demonstrates a causal link with cataract formation, indicating the need for future investigations of age-related eye diseases.

## 1. Introduction

A recent study reported that the leading causes of blindness in those aged ≥50 years in 2020 were expected to be cataract (15.2 million cases [95% uncertainty intervals (UI): 12.7–18.0]), glaucoma (3.6 million cases [95% UI: 2.8–4.4]), and age-related macular degeneration (AMD; 1.8 million cases [95% UI: 1.3–2.0]) [1,2]. As these are characteristic geriatric eye disorders, which share aging as a risk factor, glaucoma or AMD frequently occur with cataract in elderly patients; nonetheless, differences in clinical severity, treatment, and prognosis must be considered.

Glaucoma is a progressive optic neuropathy characterised by the destruction of retinal ganglion cells and their axons [3], and primary open-angle glaucoma (POAG) is the most common form of glaucoma [4]. AMD is a degenerative retinal disease; the early stages are marked by drusen and pigmentary changes and the late stages by neovascular and atrophic alterations, which are the main causes of vision loss due to AMD [5,6,7]. Cataract is recognised as a disease, which can be cured surgically in developed countries; however, blindness attributable to glaucoma or AMD is irreversible. Owing to differences in the severity of these eye diseases, numerous investigations of glaucoma and AMD have recently been conducted. These investigations, which identified the pathology and risk factors for each disease, have contributed significantly to the field of public health. In addition, because age is a key risk factor shared by these diseases, and their co-morbidities are common, cataract and glaucoma—as well as cataract and AMD—studies on the relationships between AMD, glaucoma, and cataracts may be particularly significant.

Angle-closure glaucoma leads to ischaemia, which facilitates the rapid development of cataract [8] and is associated with greater lens thickness [9]. However, few studies have explored the causal connections between cataract and POAG [10,11]. Conversely, medical or surgical treatment for glaucoma might aggravate cataract formation. Patients undergoing trabeculectomy have a higher risk of cataract development and worsening [12]; more than 50% of patients will require cataract surgery within five years of undergoing trabeculectomy [13]. Given the correlation of exposure to known risk factors, such as age, sex, smoking, metabolic disease, and ultraviolet (UV) ray exposure, with AMD and cataract, investigations of the relationships between cataract or cataract surgery and AMD may be beneficial [14,15,16,17]. A previous study reported that cataract surgery may be linked to a higher prevalence of late AMD; however, the presence of cataract may also be associated with a higher prevalence of late AMD [14]. These findings support the need for further research on the association between cataract surgery and AMD. In addition, reverse-direction association and cataract can be induced by repeated intravitreal anti-VEGF injections or focal laser therapy with verteporfin during AMD treatment [18,19].

Mendelian randomisation (MR) is a genetic epidemiological approach, which employs genetic variants associated with risk factors as instrumental variables (IVs) to evaluate their causal impact on medical conditions [20,21]. Recently, several studies using MR techniques have shown the potential causal effects of several risk factors on ophthalmic diseases, such as POAG, AMD, cataract, uveitis, and myopia [22,23,24,25,26]. It is meaningful to investigate the association of cataract with glaucoma and AMD, as these are eye diseases, which share risk factors with geriatric diseases. To date, few studies have been conducted regarding the influence of cataract on AMD and glaucoma, as well as that of AMD or glaucoma on cataract using MR technology. A recent study assessed myopia by employing MR and revealed that glaucoma was significantly associated with an increased risk of cataract development [27]. More robust findings can be achieved if multiple datasets are used. Hence, it is anticipated that a large number of samples will increase the combination of BioBank Japan (BBJ) and UK Biobank (UKB) data, Genetic Epidemiology Research in Adult Health and Aging (GERA), and the International AMD Genomics Consortium (IAMDGC), leading to reliable results.

To this end, we examined the causal impact of cataract on glaucoma or AMD, as well as glaucoma or AMD on cataract, employing a bidirectional two-sample MR using recently published summary statistics from a meta-analysis from BBJ and UKB for cataract [28], a meta-analysis from GERA and UKB for glaucoma [29], and 11 sources of data, including the IAMDGC and UKB, for AMD [30].

## 2. Materials and Methods

### 2.1. Study Design

This study protocol was approved by the institutional review boards of the Veterans Health Service Medical Center (IRB No. 2022-03-034 and IRB No. 2023-03-005); the requirement for informed consent was waived because this study was performed retrospectively. This research was carried out in accordance with the Declaration of Helsinki.

### 2.2. Data Source

Schematic plots of the analytical study design are shown in Figure 1.

To explore the causal effect of the presence of cataract on the risk of POAG and AMD in multi-ethnic populations using summary statistics from genome-wide association studies (GWASs), we selected the datasets for single-nucleotide polymorphisms (SNPs) associated with cataract (Table 1) from a meta-analysis of a multi-ethnic population (*n* = 670,603; 77,713 cases and 592,890 controls) from the BBJ and UKB [28]. The glaucoma (POAG) summary statistics were adopted from a meta-analysis (*n* = 240,302; 12,315 cases and 227,987 controls) from the GERA and UKB [31]. AMD summary statistics were adopted from a meta-analysis (*n* = 105,248; 14,034 cases and 91,214 controls) of 11 sources of data, including the IAMDGC and UKB, for AMD [30]. Table 1 presents summary statistics of the included datasets.

### 2.3. Selection of the Genetic Instrumental Variables

IVs were chosen at the genome-wide significance threshold from the SNPs associated with each exposure (*p* < 5.0 × 10^−8^). Independent genetic IVs should preferably be used to prevent duplication during counting, which would understate the standard error of the MR estimate. Independent genetic IVs were carefully selected through clumping—a process, which only retains SNP per locus based on linkage disequilibrium (LD). We trimmed these SNPs using LD (*r*^2^ = 0.001, clumping distance = 10,000 kb) to verify that each IV was independent of the others. *F*-values were utilised to assess the potency of genetic IVs. The *F*-value was calculated using the formula *F* = *R*^2^(*n* − 2)/(1 − *R*^2^), where *n* denotes the sample size, and *R*^2^ denotes the exposure variance to genetic variance ratio [32]. *F*-values larger than 10 were deemed to lack evidence of instrument bias [33]. In this study, using bidirectional MR, overlapping IVs were eliminated to strengthen the hypothesis, and LD correlation analysis was used to exclude related IVs. The LD value of two SNPs (rs61871744 (*PLEKHA1*; *ARMS2*), (Exposure: cataract, Outcome: AMD); rs3750847 (*ARMS2)*, (Exposure: AMD, Outcome: cataract)) was high (*r^2^* = 0.956); thus, these SNPs were removed for IVs.

### 2.4. Mendelian Randomisation

MR analysis was implemented under the following assumptions: (1) genetic variants were significantly associated with exposure; (2) these variants were not linked to confounding variables in the exposure–outcome relationship; and (3) they should affect the outcome solely through exposure, without the influence of the horizontal pleiotropy effect. Only the first MR assumption can be formally tested. The other two MR assumptions can be disproven and examined using various sensitivity analyses, but they cannot be proven to be true. Our principal analysis method was inverse variance weighted (IVW) MR with multiplicative random effects [33,34,35]. We employed Cochran’s Q-test to assess the heterogeneity among the SNPs in the IVW approach [33]. A *p*-value of less than 0.05 for the Cochran’s Q-test indicated the presence of heterogeneity. Horizontal pleiotropy can contribute to heterogeneity by triggering variations in the causal effects of genetic variants on the results. The highest possible efficiency of the IVW analysis is achieved when all genetic variations are compatible with the three conditions for IV assessments [36]. To account for potential issues, such as pleiotropy or invalid instrument bias, we performed various sensitivity analyses to assess the validity and robustness of the results. These included the weighted median method [37], MR-Egger regression (with or without adjustment using the simulation extrapolation (SIMEX) method) [37,38], and the MR pleiotropy residual sum and outlier (MR-PRESSO) [39] method. The weighted median method estimated the causal effect by incorporating the median of the weighted ratio estimates. The weights used are the reciprocals of the ratio estimation variances. Despite the fact that up to 50% of IVs are invalid, the weighted median approach provides consistent estimates of causality [40]. By considering a non-zero intercept, which reflects the average horizontal pleiotropic effects, and the slope, which is an estimate of the causal effect, the MR-Egger analysis offers estimates of suitable causal effects even in the presence of pleiotropic effects [37]. When the presumption of no measurement error is violated (*I*^2^ < 90%), bias can be corrected using MR-Egger regression with SIMEX [38]. The heterogeneity of the MR-Egger approach was evaluated using Rücker’s Q′ statistic tests [41]. MR-PRESSO is an extension of the IVW approach, which aims to eliminate pleiotropic outliers. The MR-PRESSO global test evaluates the existence of horizontal pleiotropy across all IVs in a single MR test. This is accomplished by comparing the measured distance of all changes from the regression line (known as the residual sum of squares) to the expected distance, assuming there is no horizontal pleiotropy. If the *p*-value for the MR-PRESSO global test is less than 0.05, the MR-PRESSO outlier test determines the presence of specific horizontal pleiotropic outlier variations. All analyses were conducted using the SIMEX and TwoSampleMR packages in R (version 3.6.3; R Core Team, Vienna, Austria).

## 3. Results

### 3.1. Genetic Instrumental Variables for the Causal Association of Cataract with Glaucoma or AMD

Twenty-eight IVs at the threshold of significance (*p* < 5.0 × 10^−8^) were chosen for cataract as exposure and glaucoma as outcome in multi-ethnic populations (Table 2). In addition, 27 IVs at the threshold of significance (*p* < 5.0 × 10^−8^) were used for MR analysis of the association of cataract as exposure and AMD as outcome. The mean F-statistics for cataract were 132.84 for glaucoma and 147.63 for AMD, which were used for MR and were more than 10, indicating a minimal likelihood of weak instrument bias, with F-values of all IVs greater than 10 (Table 2 and Appendix A). Specific details of the IVs utilised in this study are provided in Appendix A.

### 3.2. Genetic Instrumental Variables for the Causal Association of Glaucoma or AMD with Cataract

Thirty-nine IVs at the threshold of significance (*p* < 5.0 × 10^−8^) were discovered for glaucoma, and seven IVs at the threshold of significance (*p* < 5.0 × 10^−8^) were chosen for AMD (Table 2). The mean F-statistics for glaucoma and AMD were 182.81 and 169.33, respectively, which were employed for MR and were larger than 10, indicating a low likelihood of weak instrumental bias (Table 2). In addition, the F-values of all IVs were higher than 10, which indicated a low chance of weak bias (Appendix A).

### 3.3. Heterogeneity and Horizontal Pleiotropy of Instrumental Variables

To assess the quality of the IVs, the I^2^ and *p*-values for Cochran’s Q statistic were calculated using IVW, Rücker’s Q’ statistic using MR-Egger, and the MR-PRESSO global test (Table 2). Cochran’s Q-test and Rücker’s Q’ test revealed evidence of instrumental heterogeneity for cataract and glaucoma (*p* < 0.05), except for AMD on cataract. Rejecting the null hypothesis in both Cochran’s Q and Rücker’s Q’ tests for heterogeneity implies that the genetic variants might exhibit pleiotropy. However, the MR-Egger regression intercepts showed no horizontal pleiotropic effects (*p* > 0.05) in all analyses, regardless of the SIMEX adjustment (Table 2). Except that for AMD to cataract, the MR-PRESSO global test (Table 2) showed horizontal pleiotropy (*p* < 0.05), and horizontal pleiotropic outliers were identified and removed. Further information regarding the IVs and outliers exhibiting horizontal pleiotropic effect is presented in Appendix A.

### 3.4. Mendelian Randomisation for Causal Association of Cataract with Glaucoma and AMD

MR analyses using the IVW method did not reveal a significant causal association between cataract and glaucoma (MR odds ratio [OR] = 1.05, 95% confidence interval (CI):0.88–1.24, *p* = 0.595) in Figure 2A. In addition, no significant causal link was identified between cataract and glaucoma using the weighted median, MR-Egger, MR-Egger (SIMEX), and MR-PRESSO techniques (*p* = 0.194, *p* = 0.433, *p* = 0.418, *p* = 0.132, respectively) in Figure 2A.

According to previous research [42], MR-PRESSO was the most reliable analysis technique (*p* = 0.132). For AMD, MR analyses using the IVW method did not reveal a significant causal relationship between cataract and AMD (*p* = 0.534; Figure 2B). Furthermore, no significant causal relationship between cataract and AMD was observed using the weighted median MR, MR-Egger, MR-Egger (SIMEX), and MR-PRESSO techniques (*p* = 0.382, *p* = 0.313, *p* = 0.318, *p* = 0.439, respectively; Figure 2B). Previous research has reported that among these methods [42], MR-Egger (SIMEX) is the most dependable method of analysis (*p* = 0.318). The genetic connection between cataract and glaucoma and AMD for each SNP is represented by scatter plots (Figure 3).

### 3.5. Mendelian Randomisation (Reverse-Wise) for Causal Association of Glaucoma or AMD with Cataract

Glaucoma was associated with an increased risk of cataract in the IVW method (MR OR = 1.05, 95% CI: 1.02–1.08, *p* < 0.001; Figure 4). In addition, glaucoma increased the risk of cataracts, as indicated by the weighted median MR (OR = 1.05, 95% CI: 1.02–1.09, *p* = 0.002), MR-Egger (OR = 1.12, 95% CI: 1.02–1.22, *p* = 0.022), MR-Egger (SIMEX) (OR = 1.14, 95% CI: 1.03–1.27, *p* = 0.020), and MR-PRESSO (OR = 1.04, 95% CI: 1.01–1.06, *p* = 0.003) techniques, and according to previous research [42], MR-Egger (SIMEX) (OR = 1.14, 95% CI: 1.03–1.27, *p* = 0.020) was the most reliable analysis method. For the effect of AMD on cataract, a significant causal relationship between AMD and cataract development was not found in IVW (OR = 1.02, 95% CI: 0.97–1.07, *p* = 0.400), MR weighted median (OR = 1.02, 95% CI: 0.98–1.07, *p* = 0.257), MR-Egger (OR = 1.05, 95% CI: 0.97–1.14, *p* = 0.284), and MR-Egger (SIMEX) (OR = 1.05, 95% CI: 0.97–1.15, *p* = 0.284). According to a prior study [42], IVW (*p* = 0.400) is the most reliable analysis method among them. The genetic connection between glaucoma and AMD on cataract for each SNP is represented by scatter plots with weak positive correlations (Figure 5).

## 4. Discussion

Recently, the results of large GWAS datasets, such as UKB and BBJ, have been released, enabling more active genetic research [28,43]. Applications of bioinformatics approaches to genetic data can be utilised to investigate disease pathophysiology and epidemiology; our previous studies recently assessed a genetic risk model based on POAG GWAS results [44] and AMD [45] and various conditions [46,47]. Extending the genetic risk model and clinical investigations based on genetic epidemiology, it can be observed that causal interference plays a crucial role because MR is a method, which uses genetics to inform investigators of the associations in traditional observational epidemiology [48]. Existing epidemiological studies on eye diseases using MR have limitations, in that only a few cohorts contain phenotypes such as cataract, glaucoma, and AMD; thus, a combined analysis of cohorts containing an eye-specific phenotype and the utilisation of two-sample MR would be beneficial. Several MR studies on ocular diseases have identified causal interference using human genetic data [22], and lipid levels, corneal thickness, inflammation, and refractive errors have been evaluated as glaucoma risk factors [25,49,50].

Cataract, glaucoma, and AMD diseases can lead to blindness [1,2]. Exploring the associations between these diseases and the risk factors leading to blindness is anticipated to have positive implications for public health. In particular, this study investigated the potential causal effect of cataract on glaucoma and AMD, which are age-related eye diseases. Even though aging is a shared risk factor, the presence or absence of cataract could identify the cause of glaucoma or AMD. It is necessary to consider the epigenetic factors involved in these disease associations. Physicians do not modify the internal medium of the eye before phacoemulsification for the treatment of cataract. However, for the treatment of glaucoma, the standard treatment is to reduce the production of aqueous humour by modifying the internal medium using drugs or surgery; this could in turn facilitate the appearance of cataracts. In addition, for the treatment of AMD, the intraocular medium is modified with frequent intravitreal injections, which can facilitate the development of cataracts due to surgical trauma or significant changes in the levels of cytokines in the vitreous humour. Hence, this study investigated the causal associations between glaucoma and cataract, as well as AMD and cataract, as a thorough examination of these relationships was anticipated to yield more meaningful results. Based on the analysis herein, there were no causal relationships of cataract with glaucoma or AMD, or of AMD with cataract. However, a significant causal association of glaucoma with cataract formation was observed.

Cataract is defined as opacity of the lens due to loss of optical clarity concomitant with aging [51]. Cataract is well known in terms of causes, aging, medications, and exposure to UV rays; however, other metabolic factors, such as diabetes or obesity, may also be risk factors. An Asian study reported the association of an obesity-related gene (*FTO*) with cataract [52]. As cataract is affected by significant environmental components, the significance of SNP discovery is limited, e.g., when it is used as an IV for MR rather than to explain genetic predisposition. In this study, the SNPs with high OR values for cataract exposure on glaucoma were *ZNF800* (rs62621812 with OR = 1.213), *LOC338694;MYEOV* (rs79721202 with OR = 1.144), *BMP3* (rs72868578 with OR = 1.131), *SOX2-OT* (rs9823623 with OR = 1.093), and *BET1L* (rs11245997 with OR = 1.085) (Appendix A). On the other hand, the SNPs with high OR values for cataract exposure on AMD were *ZNF800* (rs62621812 with OR = 1.213), *LOC338694;MYEOV* (rs79721202 with OR = 1.144), *BMP3* (rs72868578 with OR = 1.131), *SLC24A3* (rs4814857 with OR = 1.112), and *IGFBP3;LOC730338* (rs17172647 with OR = 1.067).

UV ray exposure is a shared risk factor for AMD and cataract, and because cataract surgery (intraocular lens) increases UV ray exposure in the macular area, the prevalence of AMD may consequently increase, according to the meta-analysis results [53]. Thus, the abovementioned factors are thought to exert an influence in terms of the prevention of AMD by cataract, thereby counteracting the cataract-inducing effect of aging. Hence, the MR analysis of the causal effects of cataract on glaucoma or AMD was not significant because there were SNPs with an effect, which offset the negative direction. Therefore, it is possible to explain the association between cataract, AMD, and glaucoma in relation to aging; however, there was no causality.

Glaucoma is a common leading cause of irreversible blindness, with progressive retinal ganglion cell degeneration and unique visual field loss being the main clinical manifestations [54]. A recent study showed an association between glaucoma, cataract, and cognitive performance with respect to aging [10]. As the aging population continues to increase, dementia is a growing concern among older populations, affecting 24 million people worldwide, with rates predicted to double every 20 years [55]. Alzheimer’s disease and glaucoma share genetic risk factors, and similar pathological changes have been observed in the optic nerves and brains of glaucoma patients [56]. In this study, we aimed to analyse the causal association between cataract and glaucoma, regardless of the presence of Alzheimer’s disease. These MR analyses revealed a significant causal association between glaucoma and cataract. As we used SNPs as IVs for MR in our study, we believe it is advantageous to investigate the associated SNPs. However, this study focused on causality verification rather than genetic analysis; thus, caution is required when interpreting the results, as genome research was not the primary focus of this study. Nevertheless, our findings are consistent with those of a recent study, which showed a significant association between glaucoma and the risk of cataract development (OR = 1.04, 95% CI, 1.01–1.08) [27]. This study assessed more diverse risk factors, such as myopia, than our study and performed multi-variable MR, making the results more reliable. Of note, to reduce the effects of population stratification on our results and ensure more robust findings, we performed additional analysis for the association between cataract and glaucoma (Additional File S1). Additionally, a recent MR study revealed that type 2 diabetes has causal association with glaucoma (OR = 1.08, 95% CI, 1.02–1.13) and cataract (OR = 1.07, 95% CI, 1.03–1.11) [57]. The novelty of our study stems from utilising a distinct dataset within our study, and the substantial correlation is supported by the fact that comparable outcomes were achieved.

The reason for the causal association between glaucoma and cataract should be elucidated in two aspects: genetic aspects and environmental factors. Regarding the genetic aspect, SNPs with high OR values for glaucoma exposure on cataract were *TMCO1-AS1* (rs2814471 with OR = 1.370), *GAS7* (rs12602519 with OR = 1.178), *ABCA1;SLC44A1* (rs2472493 with OR = 1.162), and *GNB1L;TXNRD2* (rs58714937 with OR = 1.153) (Appendix A). Transmembrane and coiled-coil domains 1-antisense RNA 1 (*TMCO1-AS1*) had a significantly high OR value and was expressed in most tissues in the human eye, including the ciliary body, trabecular meshwork, and retina. This result was consistent with previous findings on cytoplasmic RNA granule function as related to cataract formation [58,59]. One study using microRNA in the aqueous humour of POAG patients showed that microRNA-122-5p related to *TMCO1* was expressed differently with the MAPK signal pathway [60]. In addition, these results support the hypothesis that the exfoliation syndrome is related to cataract and glaucoma, as demonstrated by the differential expression of microRNA-122-5p [60]. In the cataract population, sharing of these glaucoma-associated SNPs may result in an effect association. Additionally, recent advances in the field of oxidative stress and RNA binding protein using next-generation sequencing would unravel the molecular mechanisms using transcriptomic analyses [61,62,63]. Future investigations are needed to understand the shared molecular biological pathways involved in the relationship between POAG and cataract.

Next, regarding environmental factors, because the MR principle is an extension of the traditional causality verification method, it is possible to assume the existence of other variables. Patients with glaucoma typically use topical anti-glaucoma medications, such as miotic or β-blockers and prostaglandin analogues, which may cause earlier cataract formation [64]. In addition, laser treatment and trabeculectomy or minimally invasive glaucoma surgery can accelerate cataract formation. Despite a previous report indicating no difference in lens autofluorescence in patients with glaucoma or ocular hypertension (*n* = 22) compared with the control group (*n* = 24) [65], we believe that these results are inconsistent with the results of a recent study [64], as well as our study. In addition, glaucoma eye drops, such as fixed combinations, are more commonly used today than 28 years ago; thus, we believe this difference exists.

AMD, a representative disease in geriatric ophthalmology, is a well-known disease with heritable characteristics, which was the initial focus of GWAS research [66,67,68,69,70]. With these genomic data, MR-based causal studies of dyslipidaemia or other risk factors for AMD have been conducted [71,72,73,74,75,76]. In addition, the C-reactive protein is a hallmark lipid in drusen [77]. As the severity of AMD is higher than that of cataract, whether cataract surgery causes AMD has been a major focus of many studies [53,78,79,80]. Consequently, research on whether AMD causes cataract formation is limited. According to our study, the SNPs with high OR values for AMD exposure on cataract included *ARMS2* (rs3750847, age-related maculopathy susceptibility 2). However, IVs were removed during analysis owing to a high LD association value with rs61871744 (*PLEKHA1;ARMS2*) for cataract exposure on AMD. Hence, the SNPs with high OR values showing a positive association of AMD with cataract included *C3* (rs11569415 with OR 1.123), followed by *HERPUD1;CETP* (rs247617 with OR 1.097) and *RP11-1149O23.3* (rs13278062 with OR 1.083). Given that cataract is metabolically influenced, and both AMD and dyslipidaemia are metabolic diseases, it is possible that patients with AMD develop cataract. Furthermore, the use of laser therapy or intravitreal anti-VEGF injection for AMD treatment can induce cataract. However, the MR results revealed that AMD is not a causal factor for cataract development.

A notable strength of this study is the use of a large dataset, which included information on cataract, glaucoma, and AMD, revealing the causal association between glaucoma or AMD and cataract. In addition, our study employed various MR tests and a distinctive dataset (Appendix A), which ensured the robustness of our findings. Various MR procedures are influenced by the genetic architecture. However, a reliable association could be established if these results consistently yield comparable outcomes. However, this study has a few limitations. First, because we did not have access to individual-level data, we were unable to explain the presence of multiple confounding factors using summary statistics based on two-sample MR. Although MR was used to infer causality in our study, it is essential to be more cognisant of the potential pitfalls associated with population stratification. This confounding problem may arise when variations in allele frequency and disease prevalence between subpopulations produce misleading associations. To overcome this problem, additional analysis was conducted for genetic data from Europeans, and the results were similar to those of the main analysis (Appendix A). Second, the testing methods confirming MR assumptions do not completely validate the results. However, verifying that IVs do not affect the confounders and only impact the outcome through exposure remains challenging. To address this issue, we performed sensitivity studies using several MR techniques. Nevertheless, including invalid IVs in the MR analysis can introduce bias; thus, caution should be exercised when interpreting the results. Third, although cataract can impair the visibility of the retina and potentially distort visual field testing for glaucoma, it is essential to clarify how this could potentially influence our genetic association analyses. The potential of cataract to obfuscate the diagnosis of AMD and POAG is notable, but further investigation is required. Fourth, because there are few genome datasets, which include ophthalmic phenotype data, it is difficult to separate and prepare a summary of a meta-analysis, which includes a portion of the UKB. As AMD contains 11 distinct datasets, including the IAMDGC, and there are differences between the data, interpretation must be performed with caution. However, considering the research results according to the large-cohort MR analysis methodology [81], it is reported that the IVW and weighted median are not affected, which affects the bias of MR-Egger. Although UKB was partially included in the exposure and outcome GWAS in this investigation, prior research supports its reliability [81]. Additionally, because there was no substantial difference between the MR methodologies, we believe that our results are credible. Fifth, although glaucoma, AMD, and cataract commonly affect older individuals, there is a slight variation in the ages at which these diseases develop. This variability in the age of onset might lead to an underestimation of the GWAS of patients when characterising their phenotype, especially cataract, which could be a potential source of bias. While the MR method concept is not influenced by age, it is important to acknowledge the limits of the utilised dataset and exercise caution when interpreting the results.

## 5. Conclusions

We investigated the causal relationships among glaucoma, AMD, and cataract, which share the same risk factors associated with aging. Causal associations of cataract with glaucoma and AMD were not observed, in contrast to findings of previous studies, which linked cataract surgery to glaucoma or AMD. In the reverse-direction analysis, we found substantial evidence that glaucoma has a causal link with cataract, but there was no causal relationship between AMD and cataract. Through additional analysis, it was proven that glaucoma (POAG) was causally related to the development of cataracts. While the relationship was discovered through SNP-based MR analysis, it is important to also address the possibility that cataracts can be caused by the usage of eye drops for glaucoma treatment from an epigenetic standpoint. Conducting further investigations will be necessary in the future to study the biological roles of SNPs associated with the incidence of glaucoma and cataracts using transcriptome-based technologies.

## Figures and Tables

**Figure 1 genes-15-00413-f001:**
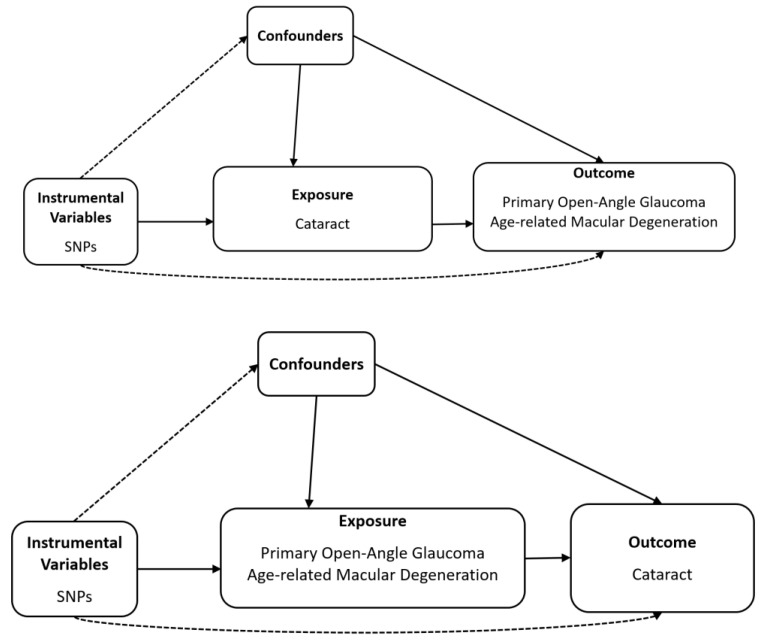
Diagram of bidirectional Mendelian randomisation analysis. Dashed lines indicate association that violate the MR assumptions, while solid lines represent observed relationship, SNP, single-nucleotide polymorphism.

**Figure 2 genes-15-00413-f002:**
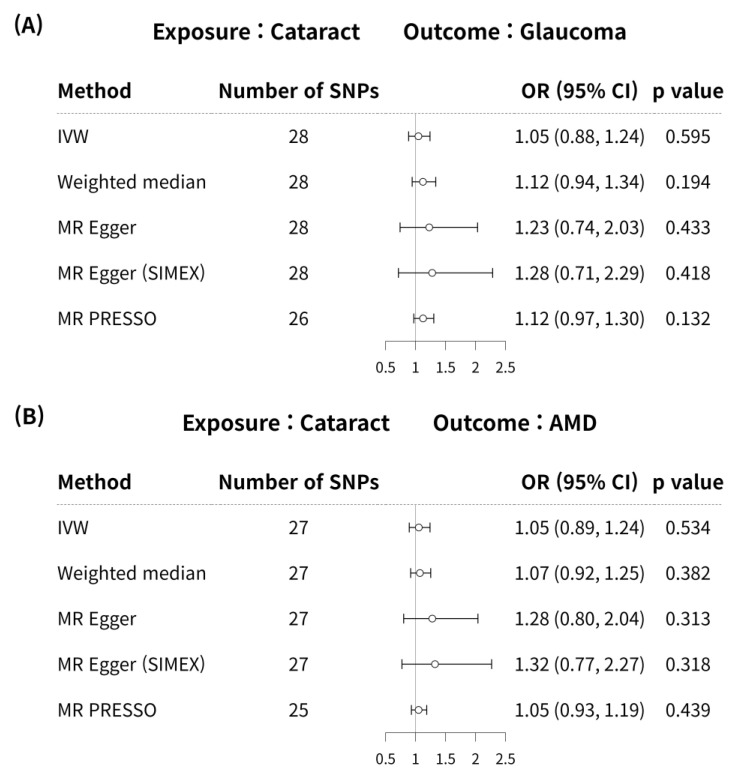
Forest plot of causal associations between cataract and glaucoma or AMD. Forest plot of causal associations between (**A**) cataract and glaucoma and (**B**) cataract and AMD. AMD, age-related macular degeneration; CI, confidence interval; SNP, single-nucleotide polymorphism; OR, odds ratio; IVW, inverse variance weighted method; MR, Mendelian randomisation; SIMEX, simulation extrapolation; MR PRESSO, MR pleiotropy residual sum and outlier.

**Figure 3 genes-15-00413-f003:**
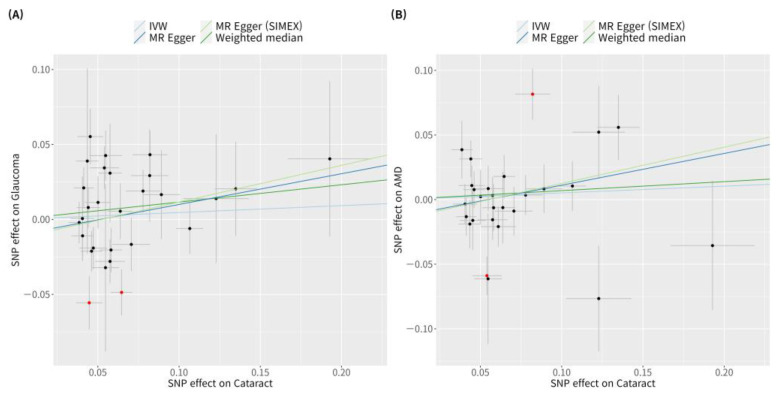
Scatter plots of MR tests assessing the effect of cataract on glaucoma or AMD. Scatter plots of MR tests assessing the effect of (**A**) cataract on glaucoma or (**B**) cataract on AMD. Light blue, light green, dark blue, and dark green regression lines represent the IVW, MR-Egger (SIMEX), MR-Egger, and weighted median estimate, respectively. Red dots indicate outliers based on the MR-PRESSO. When outliers were present, MR-PRESSO results were generated, and other MR methods were analysed without outlier removal. MR, Mendelian randomisation; SNP, single-nucleotide polymorphism; IVW, inverse variance weight; SIMEX, simulation extrapolation; AMD, age-related macular degeneration.

**Figure 4 genes-15-00413-f004:**
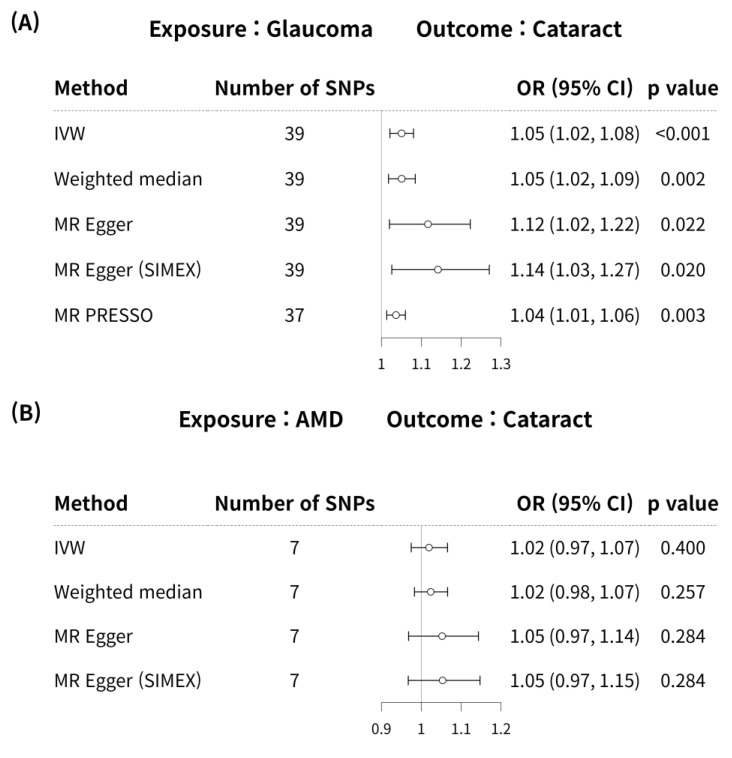
Forest plot of causal associations between glaucoma or AMD and cataract. Forest plot of causal associations between (**A**) glaucoma and cataract and (**B**) AMD and cataract. AMD, age-related macular degeneration; SNP, single-nucleotide polymorphism; OR, odds ratio; CI, confidence interval; IVW, inverse variance weight; MR, Mendelian randomisation; SIMEX, simulation extrapolation; PRESSO, pleiotropy residual sum and outlier.

**Figure 5 genes-15-00413-f005:**
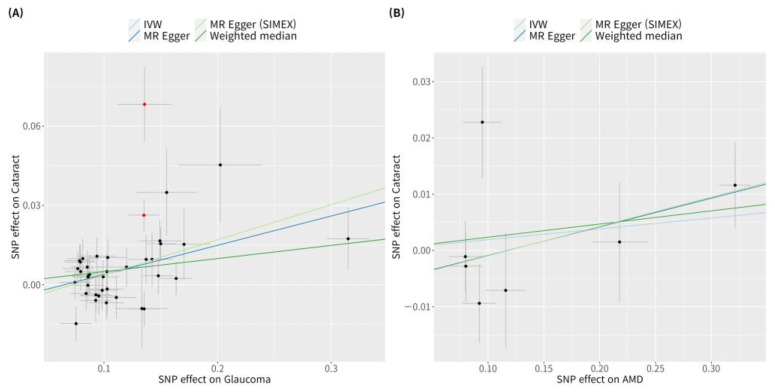
Scatter plots of MR tests assessing the effect of glaucoma or AMD on cataract. Scatter plots of MR tests assessing the effect of (**A**) glaucoma on cataract or (**B**) AMD on cataract. Light blue, dark blue, dark green, and light green regression lines represent the IVW, MR-Egger, weighted median, and MR-Egger (SIMEX) estimate, respectively. Red dots indicate outliers based on the MR-PRESSO. When outliers were present, MR-PRESSO results were generated, and other MR methods were analysed without outlier removal. SNP, single-nucleotide polymorphism; IVW, inverse variance weight; MR, Mendelian randomisation; SIMEX, simulation extrapolation; AMD, age-related macular degeneration.

**Table 1 genes-15-00413-t001:** Summary statistics of data source.

Trait	Data Source	No. of Participants	Population	No. of Variants	Reference
Cataract	BBJ + UKB	670,603 77,713 cases + 592,890 controls	East Asian + European	25,845,331	[28]
Glaucoma	GERA cohort + UKB	240,302 12,315 cases + 227,987 controls	Multi-ethnic: 214,102 European; 5103 African unspecified; 3571 Other admixed ancestry; 1847 African American or Afro-Caribbean; 5189 Hispanic or Latin American; 5370 East Asian; 5120 South Asian.	7,760,820	[29]
AMD	11 sources of data, including the IAMDGC and UKB	105,248 14,034 cases + 91,214 controls	European	11,703,383	[30]

BBJ, BioBank Japan; UKB, UK Biobank; GERA, Genetic Epidemiology Research in Adult Health and Aging; AMD, age-related macular degeneration; IAMDGC, International AMD Genomics Consortium.

**Table 2 genes-15-00413-t002:** Heterogeneity and horizontal pleiotropy of instrumental variables.

Exposure	Outcome				Heterogeneity		Horizontal Pleiotropy				
								MR-Egger		MR-Egger (SIMEX)	
		N	F	I^2^ (%)	*p* *	*p* ^†^	*p* ^‡^	Intercept, β (SE)	*p*	Intercept, β (SE)	*p*
Cataract	Glaucoma	28	132.84	94.31	<0.001	<0.001	<0.001	−0.011 (0.016)	0.517	−0.013 (0.018)	0.489
Cataract	AMD	27	147.63	89.53	<0.001	<0.001	<0.001	−0.013 (0.015)	0.393	−0.016 (0.018)	0.384
Glaucoma	Cataract	39	182.81	81.86	0.001	0.002	0.001	−0.007 (0.005)	0.173	−0.010 (0.006)	0.121
AMD	Cataract	7	169.33	97.34	0.168	0.165	0.308	−0.006 (0.007)	0.413	−0.006 (0.007)	0.409

N, number of instruments; F, F-statistic; MR, Mendelian randomisation; SIMEX, simulation extrapolation; SE, standard error; β, β coefficient; AMD, age-related macular degeneration. ***** Cochran’s Q-test from inverse variance weight. ^†^ Rücker’s Q’ test from MR-Egger. ^‡^ MR pleiotropy residual sum and outlier global test.

## Data Availability

The genome-wide association study (GWAS) datasets can be found in the GWAS catalogue (https://www.ebi.ac.uk/gwas/summary-statistics, accessed 19 July 2022).

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
