# Peer review of "Causal Associations of Glaucoma and Age-Related Macular Degeneration with Cataract: A Bidirectional Two-Sample Mendelian Randomisation Study"

_genes, 2024, doi:10.3390/genes15040413_

Round 1
Reviewer 1 Report
Comments and Suggestions for Authors
The manuscript "Causal Associations of Glaucoma and Age-Related Macular Degeneration with Cataract: A Bidirectional Two-Sample Mendelian Randomization Study" explores the genetic associations and potential causal relationships between glaucoma, age-related macular degeneration (AMD), and cataract using Mendelian Randomization (MR). Here are detailed suggestions for revision:
- Introduction and Background:
o Clarify the novelty and contribution of your study compared to existing literature. While you've identified the gap, emphasizing how your findings advance the field could strengthen the introduction.
- Methodology:
o Provide a more detailed explanation of the MR assumptions and how they are satisfied in your study context. Discuss any potential biases or limitations introduced by the selected instrumental variables (IVs).
o Elaborate on the selection criteria for SNPs used as IVs, especially regarding linkage disequilibrium and how it affects the independence of IVs.
- Results:
o Offer a deeper interpretation of the results, particularly the lack of causal effects in certain directions. Discuss the biological or clinical implications of these findings.
o Enhance the presentation of results with more detailed figures or tables that clearly show the distribution of effect sizes and confidence intervals.
- Discussion:
o Expand the discussion on the implications of your findings for clinical practice or public health policy. How might these results influence future research directions or treatment strategies?
o Address potential limitations of your study more thoroughly, including the generalizability of your findings across different populations and the implications of any residual confounding or bias.
- Statistical Analysis and Validation:
o Provide more details on the statistical methods used for sensitivity analysis, particularly how you assessed and addressed potential pleiotropy and its impact on the MR estimates.
o Discuss the robustness of your findings across different MR methods and what these results suggest about the underlying genetic architecture of these eye diseases.
- Figures and Supplementary Material:
o Ensure that all figures, including scatter plots and forest plots, are clear and adequately annotated. Consider adding supplementary figures or tables to support your findings if necessary.
o In the supplementary material, include a detailed description of the SNPs used as IVs, including their association with the exposure and outcomes, to aid in reproducibility and further analysis by readers.
- References and Citations:
- Verify that all references are current and relevant, ensuring they reflect the most recent advances in the field of oxidative stress and DED. Verify all references for accuracy and relevancy, ensuring they reflect the most current research in the field of RNV and RBPs. In particular, consider including additional references to support the discussion and to provide context to the study’s findings. I suggest adding data related to recent bulk transcriptomics studies which could represent a strong substrate to enforce the role of described molecular mechanisms, such as the recent PMID: 36490268.
- Conclusion:
o Summarize the key findings more succinctly, highlighting the clinical or genetic insights gained from your study.
o Offer specific recommendations for future research based on your findings, including potential new areas of investigation or methodologies that could further elucidate the causal relationships among these eye conditions.
Comments on the Quality of English Language
The English should be improved.
Author Response
Review #1
The manuscript "Causal Associations of Glaucoma and Age-Related Macular Degeneration with Cataract: A Bidirectional Two-Sample Mendelian Randomization Study" explores the genetic associations and potential causal relationships between glaucoma, age-related macular degeneration (AMD), and cataract using Mendelian Randomization (MR). Here are detailed suggestions for revision:
- Introduction and Background:
o Clarify the novelty and contribution of your study compared to existing literature. While you've identified the gap, emphasizing how your findings advance the field could strengthen the introduction.
Response: Thank you for your valuable comments. We revised the introduction section accordingly and incorporated the novelty of the study and its contribution to the literature (lines 71–75).
- Methodology:
o Provide a more detailed explanation of the MR assumptions and how they are satisfied in your study context. Discuss any potential biases or limitations introduced by the selected instrumental variables (IVs).
Response: Thank you for your valuable comments. We revised and provided a more detailed explanation for the MR assumptions (lines 129–136). In addition, we added in the limitations section the potential biases of the selected variables (lines 425-430, 443-450).
o Elaborate on the selection criteria for SNPs used as IVs, especially regarding linkage disequilibrium and how it affects the independence of IVs.
- Results:
o Offer a deeper interpretation of the results, particularly the lack of causal effects in certain directions. Discuss the biological or clinical implications of these findings.
Response: Thank you for your valuable comments. We revised the results section (lines 242–244) and discussed the biological and clinical implications of our findings (lines 296–305 and 349–360 356-360, 376-378).
o Enhance the presentation of results with more detailed figures or tables that clearly show the distribution of effect sizes and confidence intervals.
Response: Thank you for your valuable comments. We modified the axis of the forest plot and confidence intervals for better presentation of the results (Figures 2 and 4). In addition, we updated “additional file 1” which contains additional figures and study findings.
- Discussion:
O Expand the discussion on the implications of your findings for clinical practice or public health policy. How might these results influence future research directions or treatment strategies?
Response: Thank you for your comments that aim at enhancing the overall quality of our manuscript. We added relevant details regarding the implications of our study findings in clinical practice and public health policy in the discussion section (lines 291–293 and 296–305).
o Address potential limitations of your study more thoroughly, including the generalizability of your findings across different populations and the implications of any residual confounding or bias.
Response: Thank you for your valuable comment. We incorporated a more detailed description of the potential study limitations as suggested (lines 412–416 and 424–430).
- Statistical Analysis and Validation:
o Provide more details on the statistical methods used for sensitivity analysis, particularly how you assessed and addressed potential pleiotropy and its impact on the MR estimates.
Response: Thank you for your valuable comment. We provided more details regarding the statistical methods used for sensitivity analyses in the Methods and Results sections (lines 136–163 and 193–203).
o Discuss the robustness of your findings across different MR methods and what these results suggest about the underlying genetic architecture of these eye diseases.
Response: Thank you for your comments. As suggested, the robustness of our findings across different MR methods and datasets were discussed (Additional file 1). We added some additional comments in the discussion section for clarity (lines 412–416).
- Figures and Supplementary Material:
o Ensure that all figures, including scatter plots and forest plots, are clear and adequately annotated. Consider adding supplementary figures or tables to support your findings if necessary.
Response: Thank you for your comments. We changed the forest plot to enhance the reader understanding (Figures 2 and 4). The “Additional file 1” was updated by modifying the relevant tables and figures.
o In the supplementary material, include a detailed description of the SNPs used as IVs, including their association with the exposure and outcomes, to aid in reproducibility and further analysis by readers.
Response: Thank you for your comment. We included the beta, SE, and P values in “Supplement Table 1 as IVs” to enhance the reader understanding.
- References and Citations:
o Verify that all references are current and relevant, ensuring they reflect the most recent advances in the field of oxidative stress and DED. Verify all references for accuracy and relevancy, ensuring they reflect the most current research in the field of RNV and RBPs. In particular, consider including additional references to support the discussion and to provide context to the study’s findings. I suggest adding data related to recent bulk transcriptomics studies which could represent a strong substrate to enforce the role of described molecular mechanisms, such as the recent PMID: 36490268.
Response: Thank you for your valuable comments. We revised the relevant sections to include deeper insights from recent transcriptomics studies (line 376–378).
- Conclusion:
o Summarize the key findings more succinctly, highlighting the clinical or genetic insights gained from your study.
Response: Thank you for your comments. We summarized the key findings to highlight the clinical and genetic insights of our study (lines 457–464).
o Offer specific recommendations for future research based on your findings, including potential new areas of investigation or methodologies that could further elucidate the causal relationships among these eye conditions.
Response: Thank you for your comments. We added the future direction in the discussion section (lines 457–464).
Reviewer 2 Report
Comments and Suggestions for Authors
Review
1. Line 49-50. This paragraph can be commented, angle-closure glaucoma generates an important ischemia that facilitates the rapid development of the cataract. It seems to be little related to polymorphisms.
1.Although the mechanism of angle-closure glaucoma is associated with the advancement |
of cataract [8] and lens thickness [9]. |
|
2. Improve the edition of table 1, somewhat confusing. The information is very grouped on the left side of the table.
3. Discussion
There are some works with results similar to this: In IOVS (Cheng Jiang, 2023) and in Frontiers in endocrinology (Rumen Cheng 2023). The odds of association between glaucoma and cataract that appear in the 3 works range between 1.05-1.10 (which does not rule out associated epigenetic factors). Some of these works consider that there is a causal relationship between both pathologies. It is a novel work and provides information on studied polymorphisms.
It is necessary to comment on the epigenetic factors: In the cataract we do not modify the internal medium of the eye until we operate. However, in glaucoma we reduce the production of aqueous humor with drugs or surgery by modifying the internal medium, which facilitates the appearance of cataracts. In exudative macular degeneration we modify the intraocular medium with frequent intravitreal injections, which can facilitate the development of cataracts well due to surgical trauma or important changes in the levels of cytokines in the vitreous humor.
4. Discussion.
In developed countries, open-angle glaucoma can appear at an average age of 65-66 years, senile cataract at 74-75 years can result in cataract surgery and senile macular degeneration appears at an average of 74 to 80 years. In his work he does not refer to the average age of these population groups. He believes that the distinct age of appearance could be a confusing factor in his results?.
5. Conclusion
Consider removing this paragraph from the conclusion: In addition to the genetic correlation, it is clinically interesting that the treatment for glaucoma can induce cataract. These causal correlations between glaucoma or AMD and cataract imply that cataract formation may be related to ophthalmic diseases such as glaucoma or AMD and warrant clinical attention.
Author Response
Review 2
- Line 49-50. This paragraph can be commented, angle-closure glaucoma generates an important ischemia that facilitates the rapid development of the cataract. It seems to be little related to polymorphisms. 1.Although the mechanism of angle-closure glaucoma is associated with the advancement of cataract [8] and lens thickness [9].
Response: Thank you for your insightful comment regarding the underlying mechanisms of angle-closure glaucoma and cataract. We implemented the suggested changes in the introduction section (lines 49–50).
- Improve the edition of table 1, somewhat confusing. The information is very grouped on the left side of the table.
Response: Thank you for your comment. Table 1 was restructured for better clarity and readability.
- Discussion
There are some works with results similar to this: In IOVS (Cheng Jiang, 2023) and in Frontiers in endocrinology (Rumen Cheng 2023). The odds of association between glaucoma and cataract that appear in the 3 works range between 1.05-1.10 (which does not rule out associated epigenetic factors). Some of these works consider that there is a causal relationship between both pathologies. It is a novel work and provides information on studied polymorphisms.
It is necessary to comment on the epigenetic factors: In the cataract we do not modify the internal medium of the eye until we operate. However, in glaucoma we reduce the production of aqueous humor with drugs or surgery by modifying the internal medium, which facilitates the appearance of cataracts. In exudative macular degeneration we modify the intraocular medium with frequent intravitreal injections, which can facilitate the development of cataracts well due to surgical trauma or important changes in the levels of cytokines in the vitreous humor.
Response: Thank you for your comments which are of a significant value. The requested changes were applied in the discussion section. (lines 296–305 and 356–360).
- Discussion.
In developed countries, open-angle glaucoma can appear at an average age of 65-66 years, senile cataract at 74-75 years can result in cataract surgery and senile macular degeneration appears at an average of 74 to 80 years. In his work he does not refer to the average age of these population groups. He believes that the distinct age of appearance could be a confusing factor in his results?
Response: Thank you for your insightful comment. We incorporated this comment as a limitation in the discussion section (lines 443–450).
- Conclusion
Consider removing this paragraph from the conclusion: In addition to the genetic correlation, it is clinically interesting that the treatment for glaucoma can induce cataract. These causal correlations between glaucoma or AMD and cataract imply that cataract formation may be related to ophthalmic diseases such as glaucoma or AMD and warrant clinical attention.
Response: Thank you for your comment. We removed the indicated paragraph from the conclusion section (lines 464–467).